# Is there a link between endowment inequality and deception? – an analysis of students and chess players

Sven Grüner[1], Ilia Khassine[2]*

1 Martin Luther University Halle-Wittenberg, Faculty of Natural Sciences III, Institute of Agricultural and Nutritional Sciences, Chair of Agribusiness Management, Halle, Germany, 2 Faculty of Economics and Management Science, University Leipzig, Leipzig, Germany

* iliakhassine@gmail.com

**Data Availability Statement:** Our data and code are available online: https://osf.io/q37w8.

**Funding:** The financial support of the German Research Foundation (DFG, German Research Foundation –388911356) is acknowledged. The

## Abstract

This paper investigates experimentally the relationship between inequality in endowment and deception. Our basic design is adopted from Gneezy (2005): two players interact in a deception game. It is common knowledge that player 1 has private information about the payoffs for both players of two alternative actions. Player 1 sends a message to player 2, indicating which alternative putatively will end up in a higher payoff for player 2. The message, which can either be true or false, does not affect the payoffs of the players. Player 2 has no information about the payoffs. However, player 2 selects one of the two alternatives A or B, which is payoff-relevant for both players. Our paper adds value to the literature by extending Gneezy (2005) in two ways. First, we systematically vary the initial endowment of players 1 and 2 (common knowledge to both of them). Second, we do not limit ourselves to the standard population of university students but also recruit chess players that are not enrolled in any degree program. Doing so, we want to find out if our results remain robust over a non-standard subject population which is known to be experienced to some extent in strategic interactions. Our main findings are: (i) non-students behave more honestly than students, (ii) students are more likely to trust the opponent's message, and (iii) students and non-students behave differently to variation in initial endowment.

## 1. Introduction

Inequality can be found in most areas of life. Examples include the allocation of natural resources such as water and oil around the world. Material inequality is particularly widespread: global wealth is concentrated in the hands of a small number of people ([1, 2]). Inequality is often the starting point for conflicts in society (e.g. between different religions, gender wage gap, etc.). What are the behavioral foundations of inequality from a microeconomic point of view?

Episodic evidence suggests that the spectrum is multifaceted. Some people ignore poor people, others anonymously donate large amounts of money. Some people look up to rich people,

funder had no role in study design, data collection and analysis, decision to publish, or preparation of the manuscript.

**Competing interests:** The authors have declared that no competing interests exist.

while others become envious. While it is still mainstream to model individuals to derive utility exclusively from their own consumption, economists are increasingly recognizing that people are not only interested in their absolute but also in their relative position of wealth ([3]). There is a bunch of evidence that people compare themselves with others (e.g. [4–7]). In their model [8], assume that people are not only interested in their monetary payoffs (as purely selfish individuals would be) but also care about its distribution. They are supposed to dislike inequitable outcomes. Inequitable outcomes can arise both when individuals have less and when they do have more than others.

In their meta-analysis about experimental studies in economics, psychology, and sociology [9], find that people often refrain from telling lies. Our paper investigates the link between deception and inequality. In this realm, the question arises of whether people are more inclined to lie to poorer or richer individuals. Several authors have tackled this field of research recently. For example [10], find in their experimental studies a link between monetary incentives and upward social comparisons: people tend to cheat more if they know that close others earn more. Similarly [11], find experimental evidence for dishonest behavior if subjects are relatively disadvantaged in groups [12]. Link honest and dishonest behaviors to financial self-interest and equity concerns [13]. Analyze experimentally the norm that "one gets what one deserves" on honesty in a design where dishonesty entails income redistribution. The authors find a link between norm violation and the propensity toward dishonesty.

The subject of lying is a sensitive one, which complicates analyzing it. There are several possible ways to investigate the association between inequality and deception. This includes real data. For example [14], are investigating the distribution of true and false online messages on Twitter. The tendency to lie can also be examined with the help of questionnaires. The randomized response technique is well established in the literature for sensitive questions. However, we resort to economic experiments. Controlled experiments allow us to draw causal inferences. To study lying to other people (instead of lying to yourself in which the die experiment or a real effort task are quite common; e.g. [15, 16]), our paper adopts the basic design of [17]'s (2005) two-player cheap talk sender-receiver game. Player 1 has two options A and B. She is fully informed of the monetary consequences for herself and the opponent. Player 1 sends a message to player 2, indicating which of the two options is supposedly financially advantageous for player 2. This message can be honest or a lie. Player 2 remains uninformed about the monetary consequences associated with the payoffs. However, player 2 knows the message sent by player 1, and picks one of the two options which eventually will be played out (i.e., payoff-relevant) for both players. To analyze the link between inequality and deception, we extend the basic design of [17] to systematic variations in initial endowment. The topic of inequality has been tackled in the experimental literature with mixed findings. For example, in the realm of trust games [18], found evidence for inequality to matter, whereas [19] does not find evidence for inequality aversion. In our extended [17] design, we provide either player 1 or 2 with an initial endowment of €10 in the treatment conditions. In accordance with [8], we distinguish between monetary advantageous inequality and monetary disadvantageous inequality.

Unlike most experimental studies, we do not *only* recruit students as subjects. Students are readily available, which makes their recruitment relatively easy. They have low opportunity costs and steep learning curves. The latter is partly due to training in solving abstract problems ([20, 21]). In contrast, recruiting non-students often poses a challenge because of their higher opportunity costs. On average, they are older and therefore have more job experience. The many differences between students and non-students raise the question of the external validity of experimental studies with students: What can we reasonably learn from experimental studies with students if we are interested in the decision behavior of non-students? There is only a

limited number of studies that systematically compare students and non-students. According to [22], non-students behave as if they were more pro-social oriented. However, based on a literature review of 13 papers [23, 24], found no systematic, qualitative behavioral differences between students and non-students. Differences are attributed to the gap between the environment in the experiment and the expertise in the daily working life of the non-students. But non-students performed worse when they imported irrelevant heuristics into the experiment. Subject pool differences have been rarely analyzed in the realm of deception. One important exception is [25]. They compare students and nuns in a lying experiment and find differences between the subject pools (most notably, nuns are lying to their disadvantage in individual decision problems) [26]. Points out that the experiences of non-students are only helpful if the expertise of the professionals is relevant to the task in the lab and that the professionals also recognize that expertise is relevant.

To shed further light on possible differences between subject pools, we recruit both students and non-students. The non-students are chess players who are not enrolled in a degree program at a university. Chess players seem to be interesting for our experiment because they have training in strategic interactions. While playing chess, individuals not only have to think about objectively good moves but also form expectations about how the opponent might react to them. For example, an objectively perfect move could work out poorly in practice if it leads to variations where the opponent is an expert in. Similar to our experiment, in chess usually two people play against each other. Moreover, by reminding them that we are looking for chess players, there might be some weak form of priming ([27]). As [28] have pointed out people have different identities. The subjects may be in a competitive mood when they are reminded of their identity as a chess player (in chess there are only 3 outcomes: draw, win or lose).

The rest of the paper is organized as follows: Section 2 provides the experimental design. Section 3 describes the behavioral research questions. After presenting the approach to data analysis (Section 4), we describe the experimental subjects (Section 5) and analyze the data (Section 6). Section 7 concludes.

## 2. Experimental design

### 2.1. Basic design

The basic structure of the experiment is adopted from [17]'s (2005) 2-person deception game. The identity of the players is anonymized (for both players). It is common knowledge that player 1 has private information. The two players play against each other in a one-shot experiment. There are several reasons to carry out the experiment as a one-shot game [29]. Point out that many games are played uniquely in reality [30]. Remarks that people face many important decisions for a limited number in life (e.g. choosing a degree program, a spouse, or whether or not to buy a house). Furthermore, many entrepreneurial decisions are made irregularly in the sense that the economic framework conditions are bound to change at all times (e.g. capital restructuring, mergers). According to [31], issues such as reputation formation and signaling can be avoided through one-shot games. It also rules out learning effects and strategic behaviors (e.g. reciprocity).

*Player 1 (Sender of the message)*

Player 1 is fully informed about the payoffs for herself and her opponent (i.e., for both options A and B; see Table 1). Player 1 sends a message to player 2, indicating which option (A or B) results in a higher payoff for player 2. This message can be true or false. The message itself does not affect the payoffs of the players. In other words, it is cheap talk. The experiment consists of two decision situations. For both of them applies: Player 1 lies if she claims that

**Table 1. Payoffs for the players in both situations (from the perspective of player 1) [a].**

| Situation 1: Altruistic renunciation [b] | | | Situation 2: Costly punishment [b] | | |
|---|---|---|---|---|---|
| Option | Player 1 | Player 2 | Option | Player 1 | Player 2 |
| A | 9 | 12 | A | 6 | 15 |
| B | 10 | 3 | B | 5 | 5 |

(a) We presented situation 1 to the subjects first. We cannot exclude the possibility of order effects, which has to be analyzed in follow-up studies.

(b) We did not communicate the labels assigned to the situations to the experimental subjects. In contrast, we used the neutral framings "situation 1" and "situation 2."

option B leads to a higher payoff for player 2 than option A. Player 1 maximizes her payoffs if option B in situation 1 and option A in situation 2 were actually played. Note that for situation 1 this would be in line with a lie and for situation 2 it would be consistent with an honest message. To keep it simple, we stick to the terminology of [17] in this paper. However, we would like to point out that this dichotomy (honest vs. deception) could also be seen critically. For example, as [32] correctly points out honest messages could also be classified as deception if one has the expectation that the receiver does not follow the message.

*Player 2 (Receiver of the message)*

Player 2 has no information about the payoffs. She only knows the (possible) messages of player 1. However, player 2 picks one of the two options A or B, which is payoff-relevant for both players (common knowledge to both players).

## 2.2. Treatments

We extend [17] by considering systematic variations of the initial endowment. Altogether, we examine 3 scenarios (1 reference scenario and 2 treatment scenarios; Table 2). The experimental subjects were randomly assigned to one scenario only. To increase the statistical power of our analysis, subjects had to respond to both situations within one endowment scenario (between-subjects study design). We refrain from using a within-subjects study design (each experimental subject makes decisions in all endowment scenarios), as the subjects may activate different emotions in the treatments. Player 1 and player 2 know that the initial endowment in the respective treatments and are aware that it is common knowledge to both players.

*Short summary of the experimental design*. Player 1 is entirely informed about the payoffs associated with the two options of action and has private knowledge about the payoffs. Player 1 sends player 2 a message (either true or wrong; cheap talk) about her alleged payoffs. Player 2 only knows the possible messages from player 1 and the endowment of both players. However, the choice made by player 2 determines the payoffs of both players.

## 2.3. Subjects, incentives, and language

**2.3.1. Subjects.** We recruit both university students as well as non-students (the latter are also members of a chess club). Students are recruited via the online learning platform "StudIP" of the Martin Luther University Halle-Wittenberg, where the link to the experiment was

**Table 2. Treatment conditions (endowment scenarios).**

| | Player 1 | Player 2 |
|---|---|---|
| Benchmark (reference scenario) | €0 | €0 |
| Treatment I | €10 | €0 |
| Treatment II | €0 | €10 |

placed. The prerequisite to joining the experiment was that individuals were enrolled as students in a degree program at a university. The recruitment of the non-student chess players was carried out as follows: we contacted several chess clubs in Germany as well as people with the request to attend /advertise the study. For example, the German master (GM Niclas Huschenbeth) shared the link to the study on his social media platform. The support of Chess-Base GmbH (a German company that produces chess software and operates the Internet chess server "playchess.com") was very helpful to recruit the target number of subjects. A total of 30 individuals are recruited per treatment and population (i.e., a total of 180 of each population). The number of participants was primarily determined by the research budget.

**2.3.2 Incentives.** To increase the overall willingness to attend the experiment, ten subjects are randomly selected and awarded a show-up fee of €50. Moreover, we provided monetary incentives which were linked to decisions and a chance mechanism. A total of 20% of the subjects in the role of player 1 as well in the role of player 2 are randomly selected and paired with another subject from the same population. Subjects in the role of player 1 had to decide in two situations. We flipped a coin (i.e., $p = 0.5$) to determine which of the two decision situations were to be paid (i.e., random lottery payment technique). All amounts of money shown in the experiment correspond to the real €-values.

**2.3.4 Language.** We use neutral language (i.e., loaded terms, such as "deception" are not used).

# 3. Behavioral research questions

Research questions depend on the underlying concept of man. A rational profit maximizer is often used as a benchmark for actual human behavior. However, we would like to discuss the research questions primarily on the basis of a more comprehensive model of man. We assume that individuals do not only want to achieve high payoffs but also have non-negligible preferences about the distribution of wealth/endowment. Following [8], we assume that there are three determinants that are (potentially) relevant for the utility function of an individual. For illustration purposes, this will be presented formally (although the formula will not be used later in the paper): $U_i(x) = x_i - \alpha_i \max \{x_j - x_i, 0\} - \beta_i \max \{x_i - x_j, 0\}$, $i \neq j$, where the first term denotes the monetary payoff of player i, the second term describes monetary disadvantageous inequality, and the third term denotes monetary advantageous inequality. In other words, individuals dislike inequality. From a psychological point of view, it seems plausible to assume $\alpha > \beta$, i.e. that inequality is perceived more unpleasantly if one is in a monetarily disadvantageous situation.

In our research questions, we distinguish between *sender behavior* (i.e., the sender of the message, player 1) and *receiver behavior* (i.e., the receiver of the message, player 2).

## 3.1. Sender behavior

**3.1.1. To what extent does player 1 resort to honest behaviors in the reference scenarios?.** A rational profit maximizer favors B in situation 1 and A in situation 2 (if the opponent is assumed to follow one's message). Both options generate a monetary surplus of €1 for player 1 in the entire experiment. However, experimental studies of similar contexts indicate that individuals are willing to forego money-maximizing alternatives (for example when there are violations of social norms [33, 34]). Humans have multiple goals ([35]). These include an aversion to inequality or allocations that are perceived as unfair. The honest player 1 proposes in situation 1 an option of action which costs €1 for himself but increases the outcome for player 2 by €9 (*altruistic renunciation*). Honesty in situation 2 is associated with a message that would lead to a higher monetary outcome for both players. However, the benefit for player 1

amounts to only €1 while the other player receives a plus of €10. Player 1 may find this unjustified and decides to forego the €1 by choosing the egalitarian action option B (*costly punishment*). Since the decision of player 2 is payoff-relevant, the expectations of player 1 about whether player 2 follows the message or not plays a role. In his experimental study [17], found that slightly more than 80% of the subjects in the role of player 1 have had the expectation that the other player follows the message.

**3.1.2 How does the variation of the initial endowment affect honest messaging?.** Systematic variation of the endowment creates inequality. Following [8], we assume that inequality is perceived as unpleasant to some extent. This may influence expectations about the opponent's behavior.

I.  In treatment 1, player 1 has an initial endowment of €10; player 2 has €0. This surplus may lead to some psychological costs for player 1 due to inequality aversion or fairness preferences. As a result, player 1 is probably more willing to opt for pro-social options (compared to the reference scenario). In addition, player 1 may expect that player 2 trusts player 1 more (in a slightly more formal expression: player 1 expects that player 2 thinks that player 1 is willing to share a small fraction of the larger cake and therefore player 2 tends to follow the message of player 1). Thus, we assume that player 1 behaves more honestly than in the benchmark scenario. In other words: in situation 1, player 1 is more inclined to give up a small amount in order to prevent player 2 from being significantly worse off; in situation 2, player 1 is less inclined to propose option B (i.e., the egalitarian outcome), which is significantly monetarily detrimental to player 2.

II. In treatment 2, player 1 has an initial endowment of €0; player 2 has €10. This gap is probably perceived by player 1 as unpleasant (e.g. unjust or unfair). Player 1 might compensate this with a (compared to the benchmark scenario) reduced willingness to welcome a relatively high payoff for player 2. In other words, player 1 is more willing to lie (i.e., declare option B advantageous in both decision situations).

**3.1.3 Do non-students act *as if* they were more honest than students?.** Various studies find that non-students tend to be more pro-social than students ([20, 22, 31, 36, 37]). Greater pro-sociality towards the opponent means that player 1 increasingly falls back on honest alternatives of action: In situation 1, player 1 renounces €1 so that the opponent does not perform significantly worse; in situation 2, player 1 accepts inequality in which the opponent performs significantly better (instead of sacrificing €1 for equality).

**3.1.4 What other determinants can explain the decision-making behavior of player 1?.** The first 3 research questions dealt with the variables of expectation of the behavior of the opponent, treatment 2, treatment 3, and the population of interest (students vs. non-students). Now a bunch of other associations between the propensity to be honest and the following variables will be exploratory examined: victim sensitivity, beneficiary sensitivity, religiosity, interpersonal trust, gender, political view, age, and net income.

The perception of injustice and the reaction to injustice differs between people ([38]). We investigate the individual perceived disutility when others are undeservingly better off than one-self (*victim sensitivity*) and when oneself is better off for no reason (*beneficiary sensitivity*). The effect of *religiosity* cannot be determined unequivocally ex-ante. For example [39], present theoretical arguments for both positive and negative effects. Religiosity can promote that one is more cooperative towards other people (i.e., doing something good for others) as well as being intolerant towards people with a different background. *Interpersonal trust* matters for the performance of institutions. For example [40], describe a negative relationship between

trust and transaction costs. Experiences with other people may play a role in whether one is more pessimistic or optimistic about other people. Various studies describe differences in *gender*: [41] find that women are more egalitarian than men [42]; summarize in their literature review that women tend to be on average more risk-averse than men. With regard to the relevance of *political views* [41], "surprisingly" find no noteworthy differences between people who prefer right-wing parties and people who favor left-wing parties in terms of equality. Beyond that, humans are subject to change with *age*. This includes changes in the brain with age ([43]). Furthermore, life experiences increase with age. In addition, we examine the role of net income: the higher the income the less costly might generous behavior be.

## 3.2 Receiver behavior

**3.2.1 To what extent does player 2 trust the message from player 1 in the reference scenario?.** Player 2 only knows the (potential) messages of player 1 in the benchmark scenario. This is cheap talk and should not play a role according to rational choice theory. Nevertheless [17], found that almost 80% of those who acted in the role of player 2 followed payer 1's message. Therefore, it can be assumed that a large proportion of the subjects follows the message of player 1.

**3.2.2 How does the variation of the initial endowment affect the inclination to trust player 1?.**

I. In treatment 1, player 2 has an initial endowment of €0; player 1 has €10. Player 2 expects player 1 to be ready to give away some of the cake. In other words, player 2 expects player 1 to tend to act more honestly. Therefore, compared to the reference scenario, player 2 is more likely to follow the message of player 1.

II. In treatment 2, player 2 has an initial endowment of €10; player 1 has €0. Player 2 expects that player 1 considers the situation to be unfair and fears adverse discrimination. Therefore, player 2 is more probable (compared to the reference scenario) not to follow player 1's message (compared to the reference scenario).

**3.2.3 Do non-students rather than students tend to trust Player 1's message?.** A higher level of pro-sociality among the non-students can result in player 2 trusting the opponent more. Furthermore, it is conceivable that non-students are more willing to tolerate monetary disadvantageous inequality. Therefore, we assume that non-students follow the messages systematically more often than students do.

**3.2.4 What other determinants can explain the decision-making behavior of player 2?.** A bunch of associations between the propensity to trust player 1 and the following variables will be examined exploratory: victim sensitivity, beneficiary sensitivity, religiosity, interpersonal trust, gender, political view, age, education, and net income (for a description of the variables, see the Sender behavior section above, research question 4).

## 3.3. Outcome of bargaining

The highest outcome in terms of financial assets, defined as the sum of the individual payoffs of player 1 and player 2, can be realized when player 2 selects option A. Which scenario is most in line with Bentham's utilitarian *greatest happiness principle*? We expect player 1 to increasingly opt for option A in treatment 1 (compared to the baseline scenario) and player 2 to be inclined to follow this message. Compared to the benchmark scenario, presumably fewer subjects in the role of player 1 opt for option A in Treatment 2, but also fewer subjects trust the message. The overall effect is unclear and an empirical/experimental question. However, we

suspect that the sender's renunciation of option A is greater than the decline in the receiver's trust. In other words, the bargaining outcome would be greater for treatment 1 than for treatment 2 (Table 3).

## 4. Approach to data analysis

The institutional review board approval has been obtained by the German Association for Experimental Economic Research e.V. (No. sZXeRf5E). The design and approach to data analysis has been pre-registered (AER RCT Registry; AEARCTR-0005399).

### 4.1 Regression analysis

We deal with two primary outcome variables that depend on the role the experimental subjects have been assigned to. We are interested in whether the subjects send an honest or dishonest message if they play in the role of player 1 ("Decision player 1") and, if they are assigned to the role of player 2, whether they follow or not follow the message. It is important to consider subjects' expectations about the likely behavior of others because both preferences and beliefs matter (which is similar to public goods experiments, for example). A summary of the variables we take into consideration and a brief explanation is given in Table 4. If two or more items/questions are combined (e.g. beneficiary sensitivity) the calculation follows the procedure where the items/questions have been taken from. In the following, we take a look at our main specifications of the regression analysis. The questions/statements and their respective values are depicted in Table 4.

i. **Sender behavior.** For each decision situation, we perform a logistic regression because the dependent variable honesty is dichotomous (if yes = 1, otherwise 0). To increase the statistical power, we estimate a fully interactive model (i.e., interactions of the investigated independent variables with the population dummy variable). As coefficients of logistic regressions can only be meaningfully interpreted with respect to signs, we report marginal effects to adequately describe the effect size. We are considering the variables population (non-student, if $\bar{S} = 1$), expectation opponent version: player 1, and treatments ($T1$ = treatment 1, $T2$ = treatment 2). Furthermore, we address psychological and political control variables $X_{pp}$ (political view, interpersonal trust, religiosity, victim sensitivity, beneficiary sensitivity) as well as some other control variables $X_{other}$ (age, gender, net income). The analysis of the controls is exploratory.

$$Honest_{Sit\ A,\ B}\ (Y_i = 1)$$
$$= \beta_0 + \beta_1 \bar{S} + \beta_2 T1 + \beta_3 \bar{S}T1 + \beta_4 T2 + \beta_5 \bar{S}T2 + X_{pp}\beta + \bar{S}X_{pp}\beta + X_{other}\beta \ (1)$$
$$+ \bar{S}X_{other}\beta + u$$

$$Honest_{Sit\ A,\ B}\ (Y_i = 1)$$
$$= \beta_0 + \beta_1 \bar{S} + \beta_2 T1 + \beta_3 \bar{S}T1 + \beta_4 T2 + \beta_5 \bar{S}T2 + \beta_6 Expectation(j) \quad (2)$$
$$+ \beta_7 \bar{S}Expectation(j) + X_{pp}\beta + \bar{S}X_{pp}\beta + X_{other}\beta + \bar{S}X_{other}\beta + u$$

**Table 3. Expected bargaining outcome.**

| Treatment | Sender behavior | Receiver behavior | Σ(Player 1 + Player 2) |
|---|---|---|---|
| 1 (player 1 + €10) | Option A↑ | Trust↑ | T1 > T2 |
| 2 (player 2 + €10) | Option A↓ | Trust↓ | |

**Table 4. Summary of variables and their measurement.**

| Variable | Question / Statement | Values |
|---|---|---|
| Student | Are you enrolled as a student at a university? | Yes = 1, No = 0 (i.e., "Non-student" reverse) |
| Degree program (if Student = 1) | In which degree program are you enrolled? | List of several degree programs + option to add another one |
| Federal state | In which federal state do you live (main residence)? | Saxony-Anhalt (1), Saxony (2), Thuringia (3), Mecklenburg Western Pomerania (4), Brandenburg (5), Berlin (6), Bavaria (7), Bremen (8), Hesse (9), Hamburg (10), Baden-Württemberg (11), Lower Saxony (12), Northrhine-Westphalia (13), Rhineland Palatinate (14), Saarland (15), Schleswig Holstein (16) |
| Chess | Do you actively play chess in a club? | Yes = 1, No = 0 |
| Chess activity (if Chess = 1) | How many years have you been playing chess in a club? | #years |
| Expectation Opponent follows Version: Player 1 (sender) | How many people out of 100 do you think follow your message? | [0;100] |
| Expectation Opponent follows Version: Player 2 (receiver) | How many people out of 100 do you think have sent you an honest message? | [0;100] |
| Decision player 1 (sender) [Situation 1 and 2, respectively] | Which message do you want to send to the other player? Option A or Option B? | Message 1 (i.e., honest one) = 1; message 2 (i.e. dishonest one) = 0 |
| Decision player 2 (receiver) | How do you decide yourself? Do you follow the other player's message or do you decide differently? | 1 = Yes, I follow the message; 0 = No, I do not follow the message. |
| Political view[1] | In politics people often talk about "left" and "right" to distinguish different attitudes. If yo33u think about your own political views: Where would you place them? Please answer using the following scale. 0 means"entirely left", 10 means"entirely right". You can weigh your answers using the steps between 0 and 10. | [0 entirely left;10 entirely right] |
| Gender (Female = 1) | What is your gender? | 0 = Male, 1 = Female, 2 = Other |
| Education | Now it's about your years of education. Please add up the years of school education, training, and university education (if applicable). How many years do you have? | #years |
| Age | How old are you? | #years |
| Interpersonal trust[2] | 1) I am convinced that most people have good intentions. 2) You can't rely on anyone these days. 3) In general, people can be trusted. | ["don't agree at all"(1); "agree completely"(5)] |
| Religiosity[1] | Do you belong to a church or religious group? | Yes = 1, No = 0 |
| Victim sensitivity[3] | 1) It makes me angry when others are undeservingly better off than me. 2) It worries me when I have to work hard for things that come easily to others. | ["not at all"(1); "exactly"(6)] |
| Beneficiary sensitivity[3] | 1) I feel guilty when I am better off than others for no reason. 2) It bothers me when things come easily to me that others have to work hard for. | ["not at all"(1);"exactly"(6)] |
| Net income | Is your net income | less than €750 (= 1), €750 up to less than €1,500 (= 2), €1,500 up to less than €2,000 (= 3), €2,000 up to less than €2,500 (= 4), €2,500 up to less than €3,000 (= 5), more than €3,000 (= 6) |

SOEP-IS Group, 2018. SOEP-IS 2014 –Questionnaire for the SOEP Innovation Sample (Boost Sample, Update soep.is.2016.1). SOEP Survey Papers 518: Series A–Survey Instruments (Erhebungsinstrumente). Berlin: DIW Berlin/SOEP.

Beierlein, C., Kemper, C., Kovaleva, A.J. Rammstedt, B. (2014): Interpersonales Vertrauen (KUSIV3). Zusammenstellung sozialwissenschaftlicher Items und Skalen. doi: 10.6102/zis37 [English version: https://www.gesis.org/fileadmin/_migrated/content_uploads/KUSIV3_en.pdf]

Schmitt, M., Baumert, A., Gollwitzer, M. Maes, J. (2010): The Justice Sensitivity Inventory: Factorial validity, location in the personality facet space, demographic pattern, and normative data. Social Justice Research 23: 211–238. [We use the following short scale: https://zis.gesis.org/skala/Beierlein-Baumert-Schmitt-Kemper-Kovaleva-Rammstedt-Ungerechtigkeitssensibili%C3%A4t-Skalen-8-(USS-8)].

ii. **Receiver behavior.** The regressions differ from player 1 above only in the dependent variable (trust) and in the independent variable (expectations about player 1 instead of player 2):

$$Trust_{Sit\ A,\ B}\ (Y = 1)$$
$$= \beta_0 + \beta_1 \bar{S} + \beta_2 T1 + \beta_3 \bar{S} T1 + \beta_4 T2 + \beta_5 \bar{S} T2 + X_{pp}\beta + \bar{S}X_{pp}\beta + X_{other}\beta \quad (3)$$
$$+ \bar{S}X_{other}\beta + u$$

$$Trust_{Sit\ A,\ B}\ (Y_j = 1)$$
$$= \beta_0 + \beta_1 \bar{S} + \beta_2 T1 + \beta_3 \bar{S} T1 + \beta_4 T2 + \beta_5 \bar{S} T2 + \beta_6 Expectation(i) \quad (4)$$
$$+ \beta_7 \bar{S} Expectation(i) + X_{pp}\beta + \bar{S}X_{pp}\beta + X_{other}\beta + \bar{S}X_{other}\beta + u$$

iii. **Outcome of bargaining.** The decision of player 2 determines the payoffs of both players. In both situations, a monetary superior bargaining result could be achieved if option A would have been chosen. Thus, the number of A-outcomes is compared among the three scenarios and both populations. Cramér's V is used to statistically analyze dichotomous decisions.

## 4.2 Comment on p-values

There is an intensive debate and discussion on how to use and interpret p-values ([44]). Since this article is not the appropriate place to pursue the discussion in detail, we want to communicate only a few thoughts. While in the past it was quite common to focus on "statistically significant" results, the dichotomy of significant/non-significant is increasingly viewed critically. For example [44], argue "Don't believe that an association or effect is absent just because it was not statistically significant." or "In sum, "statistically significant"—don't say it and don't use it."

Notice, in our sample, there are many interaction terms (which deflates p-values) and several variables we looked at ("multiple testing", which inflates p-values). Therefore, p-values should be cautiously interpreted. The signs and strength of evidence in terms of marginal effects or differences in mean are more meaningful than just looking at p-values.

## 5. Description of the sample

As pre-registered, the sample comprises 360 subjects (half of which are enrolled in a university degree program and the other half are non-students who are members of a chess club). The size of the sample was primarily driven by budget constraints. It should be noted that there are 11 subjects among the students who also play chess in a club. The average membership of chess players in a club amounts to 27.65 years (SD = 15.06). The vast majority of the students indicated to have their main residence in Saxony-Anhalt (77.22%); the second-highest fraction of participants is from Saxony (7.22%) followed by Schleswig Holstein (3.89%). The residence of the non-students is more widespread across the various federal states: The largest fraction is from Saxony (17.78%), followed by Baden-Württemberg, and Northrhine-Westphalia (both 12.78%). As Table 5 indicates, there are considerable differences but also similarities between the two populations. Among the non-student chess players, 90% associated themselves as male. In contrast, the majority of students is female (63.33%). The fraction of the third gender is very low for both populations (<2%). Due to the low number of the third gender, we stick to the women-men-dichotomy. There is a clear gap in age: non-students are on average

**Table 5. Description of the subjects (N = 360).**

|  |  | Non-students | | Students | | Difference | |
|---|---|---|---|---|---|---|---|
|  |  | **Mean/Fraction** | **Std. Dev.** | **Mean/Fraction** | **Std. Dev.** | **Mean/Fraction** | **Std. Dev.** |
| Gender | Male | 90.00 | - | 35.00 | - | 55.00 | - |
|  | Female | 9.44 | - | 63.33 | - | -53.89 | - |
|  | Other | 0.56 | - | 1.67 | - | -1.11 | - |
| Age | | 47.33 | 14.59 | 23.40 | 3.23 | 23.93 | 11.35 |
| Political view | | 3.78 | 2.00 | 3.32 | 1.70 | 0.45 | 0.30 |
| Interpersonal trust | | 3.71 | 0.68 | 3.54 | 0.73 | 0.16 | -0.04 |
| Religiosity | | 0.30 | - | 0.35 | - | -0.05 | - |
| Victim sensitivity | | 2.78 | 1.08 | 3.52 | 1.19 | -0.73 | -0.10 |
| Beneficiary sensitivity | | 2.54 | 1.17 | 3.27 | 1.15 | -0.73 | 0.02 |
| Net income | | 4.16 | 1.58 | 1.31 | 0.53 | 2.85 | 1.05 |

considerably older (47.3 years) than students (23.4 years). Moreover, the range of age is broader among non-students than for students (18–33 years and 18–79 years, respectively). On average, non-students (M = 3.78) state their political view to be somewhat more "right" than students (M = 3.32). Students seem to trust other people slightly less (M = 3.54) than non-students (M = 3.71). When asked about belonging to a church or religious group, 30% of non-students and 35.5% of students answered yes. Victim sensitivity, as well as beneficiary sensitivity, is more pronounced, on average, for students than for non-students. As expected, the average net income of non-students (M = 4.16) is substantially higher than that of students (1.31). However, the average number of years of education is quite similar between the populations of students (M = 16.45; SD = 2.84) and non-student chess players (M = 17.63; SD = 3.20).

# 6. Results

## 6.1 Behavior of the sender

Table 6 summarizes the decision behavior of the subjects in the role of player 1. Being honest is not in line with a rational money maximizer in situation 1; the opposite applies to situation 2. The willingness to send an honest message is above 75% in the baseline scenario. Interestingly, in both situations, the non-students were slightly more honest than the students. However, the difference is small with 3.33 percentage points in situation 1 (V = -0.0405); in situation 2 it is slightly larger with 10 percentage points 2 (V = -0.1292). This observation can be explained by a higher propensity of non-students to expect the opponent to follow the message in the baseline scenario. However, the subjects in our study were less optimistic than the

**Table 6. Decisions and expectations of player 1 (sender).**

|  |  | Decision | | Expectation |
|---|---|---|---|---|
|  |  | **Honest message in situation 1** | **Honest message in situation 2** | **Expectation opponent follows** |
| Students | Baseline | 76.67 | 76.67 | M = 66.433, SD = 18.576 |
|  | T1 | 93.33 | 76.67 | M = 62.000, SD = 15.761 |
|  | T2 | 66.67 | 70.00 | M = 60.533, SD = 23.748 |
| Non-students | Baseline | 80.00 | 86.67 | M = 72.466, SD = 18.830 |
|  | T1 | 66.67 | 70.00 | M = 60.433, SD = 20.730 |
|  | T2 | 76.67 | 80.00 | M = 72.466, SD = 19.609 |

1) Honest message means that the message "Option A makes you earn more money" was sent.

subjects in [17] who found 82% of the subjects to expect player 2 to follow their message. In our study, only 66.4% (72.4%) of the students (non-students) expect player 2 to follow her message. The correlations between decisions (whether or not to send an honest message) and expectations (that the opponent follows the message) are smaller than we had expected a priori. The correlations are very weak and weak in situation 1; in situation 2 there are also moderate correlations (see Appendix, point-biserial correlation coefficient, player 2). Thus, it seems that there are other variables that might have more explanatory power than the expectation. This is addressed in the regression analysis.

In treatment 1, player 1 has an initial endowment of €10 (player 2 has €0), which is why we assumed that player 1 is more inclined to send an honest message to her opponent compared to the benchmark scenario. The expected influence is partially evident among the students. They sent considerably more honest messages in situation 1 (V = 0.2334), whereas no differences can be found in situation 2 compared to the baseline scenario (V = 0.0000). Somewhat surprisingly, non-students did less often send honest messages compared to the baseline scenario (situation 1: V = -0.1508; situation 2: V = -0.2023). Taking a look at the expectations indicates that the behavior of the non-students might be associated with a lower belief that the opponent will follow the message. Since player 1 has an initial endowment of €0 (player 2 has €10) in treatment 2, we assumed that player 2 is less likely to send an honest message compared to the benchmark scenario. In line with that we found that compared to the benchmark scenario, fewer subjects have sent an honest message. The effect is small but seems more pronounced among the students (situation 1: V = -0.1110; situation 2: V = -0.0754) than the non-students (situation 1: V = -0.0405; situation 2: V = -0.0894).

In the following, (for robustness purposes) we take a brief look at the logistic regressions to explain the tendency to send an honest message (Table 7A and 7B). The regression results are by and large in line with what we have found so far. Non-students are more inclined to send an honest message (an exception is specification IIb of Table 7A, where age and expectations were controlled.). The tendency to send an honest message is lower among non-students in treatment 1 than among students. In contrast, the decrease in treatment 2 is relatively pronounced among students, whereas little effect can be found among the non-students.

## 6.2 Behavior of the receiver

Similar to [17], we find that the majority of subjects in the role of player 2 follow player 1's message (Table 8). However, in the baseline scenario, there is a gap between students and non-students: whilst only 66.67% of the non-students follow the message of player 1, 80.00% of the students do so (cf., Table 9). This association is small according to Cramer's V (V = 0.1508). If the opponent has an initial endowment of €10 (i.e., treatment 1), the behavior of both populations deviates only slightly from the baseline scenario. The fraction of students that follow the message from player 1 is a little bit lower than in the baseline scenario (V = -0.0788), whereas the opposite is the case for non-students (V = 0.0358). It should also be mentioned that receivers, who have less initial endowment than their opponents, might not trust the senders because the former could believe that the latter may try to even the payoffs as a fairness criterion. In treatment 2, where player 2 has an initial endowment of €10, both students (V = -0.2182) and non-students (V = -0.1361) follow the message of the opponent much less compared to the respective baseline scenarios. Note that the decision-making behavior of students and non-students in treatment 1 is almost indistinguishable from perfect independence (V = -0.0370). A similar correlation can be found for students and non-students in treatment 2 (V = -0.0673). Since player 2 is only informed about the message sent by her opponent and the initial endowment of both players, it seems plausible that expectations about the likely behavior of the

**Table 7.** a Regressions to explain honest behaviors. b Regressions to explain honest behaviors.

**a**

| Logit (Marginal effects) Y = 1, message honest Y = 0, else | Situation 1 | | | | | | | |
|---|---|---|---|---|---|---|---|---|
| | Ia | | IIa | | Ib | | IIb | |
| | dy/dx (Std. Err.) | P>\|z\| | dy/dx (Std. Err.) | P>\|z\| | dy/dx (Std. Err.) | P>\|z\| | dy/dx (Std. Err.) | P>\|z\| |
| Non-student | 0.3869 (0.4548) | 0.395 | 0.2577 (0.4721) | 0.585 | 0.1383 (0.6000) | 0.818 | -0.1038 (0.5972) | 0.862 |
| Treatment 1 | 0.1438 (0.1035) | 0.165 | 0.1340 (0.1037) | 0.196 | 0.1447 (0.0995) | 0.146 | 0.1328 (0.0962) | 0.168 |
| Treatment 1 · Non-student | -0.3891 (0.2613) | 0.137 | -0.3717 (0.2649) | 0.160 | -0.3695 (0.2682) | 0.168 | -0.3357 (0.2678) | 0.210 |
| Treatment 2 | -0.1593 (0.1334) | 0.232 | -0.1682 (0.1342) | 0.210 | -0.1562 (0.1306) | 0.231 | -0.1658 (0.1293) | 0.200 |
| Treatment 2 · Non-student | 0.0910 (0.0990) | 0.358 | 0.0953 (0.0942) | 0.312 | 0.1018 (0.0863) | 0.238 | 0.1052 (0.0782) | 0.179 |
| Expectation | | | -0.0024 (0.0023) | 0.300 | | | -0.0027 (0.0022) | 0.219 |
| Expectation · Non-student | | | 0.0023 (0.0029) | 0.422 | | | 0.0033 (0.0027) | 0.232 |
| Age | | | | | -0.0213 (0.0159) | 0.179 | -0.0235 (0.0156) | 0.131 |
| Age · Non-student | | | | | 0.0158 (0.0162) | 0.329 | 0.0181 (0.0159) | 0.257 |
| Female | 0.1694 (0.0835) | 0.043 | 0.1775 (0.0809) | 0.028 | 0.1223 (0.0885) | 0.167 | 0.1314 (0.0819) | 0.109 |
| Female · Non-student | -0.1837 (0.2598) | 0.480 | -0.2038 (0.2647) | 0.441 | -0.1697 (0.2644) | 0.521 | -0.1963 (0.2706) | 0.468 |
| Political view | -0.0493 (0.0290) | 0.090 | -0.0550 (0.0290) | 0.059 | -0.0551 (0.0283) | 0.052 | -0.0633 (0.0283) | 0.025 |
| Political view · Non-student | 0.0503 (0.0346) | 0.146 | 0.0562 (0.0345) | 0.104 | 0.0498 (0.0340) | 0.143 | 0.0574 (0.0336) | 0.088 |
| Religiosity | 0.1762 (0.0893) | 0.048 | 0.1796 (0.0874) | 0.040 | 0.1853 (0.0851) | 0.029 | 0.1946 (0.0820) | 0.018 |
| Religiosity · Non-student | -0.1953 (0.2168) | 0.368 | -0.2069 (0.2202) | 0.347 | -0.2781 (0.2380) | 0.243 | -0.3122 (0.2471) | 0.207 |
| Net income | 0.1695 (0.1022) | 0.097 | 0.2031 (0.1129) | 0.072 | 0.2162 (0.1045) | 0.039 | 0.2652 (0.1165) | 0.023 |
| Net income · Non-student | -0.1734 (0.1048) | 0.098 | -0.2071 (0.1151) | 0.072 | -0.2039 (0.1076) | 0.058 | -0.2516 (0.1197) | 0.036 |
| Trust | 0.0195 (0.0660) | 0.767 | 0.0256 (.0667) | 0.701 | 0.0055 (0.0640) | 0.931 | 0.0072 (0.0630) | 0.909 |
| Trust · Non-student | -0.0283 (0.0887) | 0.750 | -0.0338 (0.0888) | 0.703 | -0.0264 (0.0855) | 0.757 | -0.0296 (0.0837) | 0.723 |
| Victim sensitivity | 0.1038 (0.0430) | 0.016 | 0.0940 (0.0439) | 0.032 | 0.0964 (0.0409) | 0.018 | 0.0845 (0.0408) | 0.038 |
| Victim sensitivity · Non-student | -0.1006 (0.0553) | 0.069 | -0.0907 (0.0555) | 0.102 | -0.1085 (0.0536) | 0.043 | -0.0971 (0.0526) | 0.065 |
| Beneficiary sensitivity | -0.0393 (0.0403) | 0.330 | -0.0343 (0.0406) | 0.398 | -0.0291 (0.0391) | 0.457 | -0.0223 (0.0384) | 0.561 |
| Beneficiary sensitivity · Non-student | 0.1069 (0.0566) | 0.059 | 0.1005 (0.0566) | 0.076 | 0.0994 (0.0554) | 0.073 | 0.0901 (0.0545) | 0.099 |
| Prob > chi2 | 0.0247 | | 0.0362 | | 0.0122 | | 0.0161 | |
| Pseudo R2 | 0.1712 | | 0.1770 | | 0.1988 | | 0.2072 | |

**b**

| Logit (Marginal effects) Y = 1, message honest Y = 0, else | Situation 2 | | | | | | | |
|---|---|---|---|---|---|---|---|---|
| | Ia | | IIa | | Ib | | IIb | |
| | dy/dx (Std. Err.) | P>\|z\| | dy/dx (Std. Err.) | P>\|z\| | dy/dx (Std. Err.) | P>\|z\| | dy/dx (Std. Err.) | P>\|z\| |
| Non-student | 0.2650 (0.5199) | 0.610 | 0.3611 (0.5379) | 0.502 | 0.9481 (0.1186) | 0.000 | 0.9499 (0.1167) | 0.000 |
| Treatment 1 | 0.0260 (0.1058) | 0.805 | 0.0348 (0.1031) | 0.735 | 0.0524 (0.1019) | 0.607 | 0.0546 (0.1001) | 0.586 |
| Treatment 1 · Non-student | -0.2849 (0.2229) | 0.201 | -0.2486 (0.2247) | 0.269 | -0.2942 (0.2263) | 0.194 | -0.2302 (0.2262) | 0.309 |
| Treatment 2 | -0.0787 (0.1261) | 0.532 | -0.0777 (0.1253) | 0.535 | -0.0667 (0.1232) | 0.588 | -0.0636 (0.1231) | 0.605 |
| Treatment 2 · Non-student | -0.0310 (0.1834) | 0.866 | -0.0207 (0.1768) | 0.906 | -0.0203 (0.1764) | 0.908 | -0.0057 (0.1689) | 0.973 |
| Expectation | | | 0.0040 (0.0024) | 0.093 | | | 0.0035 (0.0023) | 0.130 |
| Expectation · Non-student | | | -0.0017 (0.0032) | 0.584 | | | -0.0006 (0.0032) | 0.836 |
| Age | | | | | 0.0341 (0.0202) | 0.091 | 0.0315 (0.0203) | 0.122 |
| Age · Non-student | | | | | -0.0386 (0.0204) | 0.059 | -0.0370 (0.0206) | 0.073 |
| Female | -0.1714 (0.1222) | 0.161 | -0.1999 (0.1271) | 0.116 | -0.0774 (0.1239) | 0.532 | -0.1066 (0.1300) | 0.412 |
| Female · Non-student | 0.1301 (0.0918) | 0.157 | 0.1354 (0.0837) | 0.106 | 0.0503 (0.1558) | 0.747 | 0.0549 (0.1524) | 0.719 |
| Political view | -0.0139 (0.0286) | 0.627 | -0.0026 (0.0291) | 0.928 | 0.0019 (0.0300) | 0.948 | 0.0086 (0.0297) | 0.772 |
| Political view · Non-student | 0.0507 (0.0380) | 0.182 | 0.0353 (0.0384) | 0.357 | 0.0290 (0.0392) | 0.460 | 0.0166 (0.0391) | 0.671 |
| Religiosity | -0.1772 (0.1121) | 0.114 | -0.2034 (0.1191) | 0.088 | -0.2051 (0.1168) | 0.079 | -0.2201 (0.1212) | 0.069 |
| Religiosity · Non-student | 0.1570 (0.0824) | 0.057 | 0.1737 (0.0752) | 0.021 | 0.1475 (0.0857) | 0.085 | 0.1553 (0.0819) | 0.058 |

*(Continued)*

**Table 7.** (Continued)

| | | | | | | | | |
|---|---|---|---|---|---|---|---|---|
| Net income | -0.0881 (0.0805) | 0.274 | -0.0914 (0.0804) | 0.256 | -0.1518 (0.0895) | 0.090 | -0.1441 (0.0887) | 0.105 |
| Net income · Non-student | 0.0702 (0.0860) | 0.415 | 0.0779 (0.0857) | 0.363 | 0.1478 (0.0952) | 0.120 | 0.1492 (0.0944) | 0.114 |
| Trust | 0.1258 (0.0697) | 0.071 | 0.1028 (0.0692) | 0.138 | 0.1463 (0.0721) | 0.043 | 0.1164 (0.0719) | 0.105 |
| Trust · Non-student | -0.1676 (0.1073) | 0.118 | -0.1545 (0.1066) | 0.147 | -0.1950 (0.1071) | 0.069 | -0.1802 (0.1070) | 0.092 |
| Victim sensitivity | -0.0570 (0.0421) | 0.176 | -0.0502 (0.0422) | 0.234 | -0.0548 (0.0420) | 0.192 | -0.0500 (0.0423) | 0.238 |
| Victim sensitivity · Non-student | 0.0993 (0.0647) | 0.125 | 0.0891 (0.0645) | 0.167 | 0.0866 (0.0650) | 0.183 | 0.0765 (0.0653) | 0.241 |
| Beneficiary sensitivity | 0.0672 (0.0422) | 0.112 | 0.0737 (0.0427) | 0.085 | 0.0663 (0.0418) | 0.113 | 0.0738 (0.0430) | 0.086 |
| Beneficiary sensitivity · Non-student | -0.0849 (0.0633) | 0.180 | -0.0914 (0.0634) | 0.149 | -0.0829 (0.0625) | 0.184 | -0.0910 (0.0637) | 0.153 |
| Prob > chi2 | 0.3161 | | 0.2326 | | 0.2080 | | 0.1481 | |
| Pseudo R2 | 0.1099 | | 0.1303 | | 0.1335 | | 0.1545 | |

opponent are crucial. The correlations between the decisions (following player 1's message) and the expectations are middle to strong (see Appendix, point-biserial correlation coefficient, player 2).

The regression analysis to explain trust behavior provides some interesting insights (Table 9). Specification 1a shows that the dummy non-student is negatively associated with the tendency to follow the opponent's message. The comparison of specification 1a and 1b indicates that this effect is reversed when controlling for the variable age (specifications 1b to 1d). Moreover, the regressions show a strong effect for treatment 2. The control religiosity has a substantial, positive effect which is, however, only positive for the students.

## 6.3 Outcome of bargaining (welfare analysis)

To maximize the sum of the payoffs of both players, it would be best to play option A as often as possible (cf., Section 2). In situation 1, option A (9+12) exceeds option B (10+3) by €8; option A (6+15) exceeds option B (5+5) by €11 in situation 2. The variable of interest is how often option A has been played. Welfare analysis requires the two variables "Honest message" (H) and "Following message" (F). The fraction of expected A-outcomes (i.e., realized honest option) can be calculated by $EA = H \cdot F + (1–H) \cdot (1–F)$. Let us assume, for example, that $H = 0.7667$ and $F = 0.8$ is given. Thus, the expected fraction of A outcomes equals 61.336 + 4.666 = 66.002. The expected bargaining outcomes of our experimental study are summarized in Table 10.

The main findings can be summarized as follows:

1. Treatment 1 (where player 1 has an initial endowment of €10, whereas player 2 has €0) results in a better bargaining outcome than treatment 2 (where player 1 has an initial

**Table 8. Decisions and expectations of player 2 (receiver).**

| | | Decision | Expectation |
|---|---|---|---|
| | | Follow message of player 1 (= 1) | Expectation opponent sends honest message |
| Students | Baseline | 80.00 | M = 63.366, SD = 20.595 |
| | T1 | 73.33 | M = 49.933, SD = 23.648 |
| | T2 | 60.00 | M = 58.533, SD = 18.303 |
| Non-students | Baseline | 66.67 | M = 56.600, SD = 24.074 |
| | T1 | 70.00 | M = 60.433, SD = 19.890 |
| | T2 | 53.33 | M = 49.800, SD = 24.688 |

**Table 9. Regressions to explain trust behaviors.**

| Logit (Marginal effects) Y = 1, trust message Y = 0, else | Ia | | Ib | | Ic | | Id | |
|---|---|---|---|---|---|---|---|---|
| | dy/dx (Std. Err.) | P>\|z\| | dy/dx (Std. Err.) | P>\|z\| | dy/dx (Std. Err.) | P>\|z\| | dy/dx (Std. Err.) | P>\|z\| |
| Non-student | -0.1286 (0.2811) | 0.647 | 0.1146 (0.5072) | 0.821 | 0.5801 (0.6007) | 0.334 | 0.7097 (0.5859) | 0.226 |
| Treatment 1 | 0.1242 (0.1226) | 0.311 | 0.1208 (0.1226) | 0.325 | 0.1207 (0.1473) | 0.412 | 0.2793 (0.1178) | 0.018 |
| Treatment 1 · Non-student | -0.2239 (0.2436) | 0.358 | -0.2021 (0.2427) | 0.405 | -0.1278 (0.2323) | 0.582 | -0.4110 (0.3107) | 0.186 |
| Treatment 2 | -0.2268 (0.1497) | 0.130 | -0.2245 (0.1487) | 0.131 | -0.1010 (0.1477) | 0.494 | -0.1090 (0.1492) | 0.465 |
| Treatment 2 · Non-student | 0.1440 (0.1255) | 0.251 | 0.1443 (0.1227) | 0.239 | -0.0450 (0.1946) | 0.817 | 0.0853 (0.1365) | 0.532 |
| Expectation | 0.0153 (0.0031) | 0.000 | 0.0151 (0.0031) | 0.000 | - | | 0.0134 (0.0032) | 0.000 |
| Expectation · Non-student | 0.0009 (0.0050) | 0.849 | 0.0009 0.0050) | 0.853 | - | | 0.0042 (0.0057) | 0.459 |
| Age | | | -0.0012 (0.0157) | 0.937 | 0.0016 (0.0160) | 0.919 | 0.0067 (0.0158) | 0.672 |
| Age · Non-student | | | -0.0041 (0.0161) | 0.800 | -0.0056 (0.0164) | 0.732 | -0.0114 (0.0163) | 0.484 |
| Female | | | | | 0.0851 (0.1139) | 0.455 | 0.0541 (0.1136) | 0.634 |
| Female · Non-student | | | | | -0.2561 (0.24146) | 0.289 | -0.2285 (0.4162) | 0.583 |
| Political view | | | | | -0.0001 (0.0372) | 0.997 | -0.0028 (0.0355) | 0.937 |
| Political view · Non-student | | | | | -0.0248 (0.0453) | 0.583 | -0.0461 (0.0462) | 0.318 |
| Religiosity | | | | | 0.2055 (0.1066) | 0.054 | 0.1651 (0.0952) | 0.083 |
| Religiosity · Non-student | | | | | -0.3605 (0.2065) | 0.081 | -0.2342 (0.2612) | 0.370 |
| Net income | | | | | -0.1419 (0.1150) | 0.217 | -0.0430 (0.1075) | 0.689 |
| Net income · Non-student | | | | | 0.1595 (0.1191) | 0.180 | 0.0803 (0.1137) | 0.480 |
| Trust | | | | | 0.2427 (0.0824) | 0.003 | 0.1815 (0.0733) | 0.013 |
| Trust · Non-student | | | | | -0.1089 (0.1116) | 0.329 | -0.1187 (0.0997) | 0.234 |
| Victim sensitivity | | | | | -0.0241 (0.0501) | 0.630 | -0.0372 (0.0492) | 0.449 |
| Victim sensitivity · Non-student | | | | | -0.0075 (0.0715) | 0.916 | -0.0524 (0.0786) | 0.505 |
| Beneficiary sensitivity | | | | | 0.0611 (0.0546) | 0.263 | 0.0516 (0.0468) | 0.270 |
| Beneficiary sensitivity · Non-student | | | | | -0.0208 (0.0697) | 0.765 | 0.0067 (0.0653) | 0.918 |
| Prob > chi2 | 0.0000 | | 0.0000 | | 0.0233 | | 0.0000 | |
| Pseudo R2 | 0.3438 | | 0.3534 | | 0.1581 | | 0.4436 | |

endowment of €0, whereas player 2 has €10), regardless of the population. The distribution of the initial endowment appears to be non-allocation-neutral.

2. At the aggregate level, students and non-students earn less money in treatment 2 than in the baseline scenario.

## 7. Conclusion

The paper addressed the behavioral influence of differences in initial endowment on the tendency to send an honest message and to trust others. It also dealt with the question of whether

**Table 10. Expected fraction of A's (payoff-superior outcome).**

| | | Expected A-realizations in situation 1 | Expected A-realizations in situation 2 |
|---|---|---|---|
| Students | Baseline | 66.002 | 66.002 |
| | T1 | 70.217 | 62.444 |
| | T2 | 53.334 | 54.000 |
| Non-students | Baseline | 60.002 | 62.225 |
| | T1 | 56.668 | 58.000 |
| | T2 | 51.776 | 51.998 |

students and non-students differ in their behaviors. For this purpose, we adopted the basic design of [17]'s (2005) two-player deception game and extended it to two points: differences in the initial endowment and different subject pools (students and non-student chess players). The non-students are, on average, much older, earn more money, and have a systematically different gender distribution. Overall, students can be described as quite homogeneous, and non-student chess players rather heterogeneous with spite to their personal characteristics. Can different behavior patterns be observed in the experiment between the two populations? Reminding the subjects of their identity might be a driver for differences.

We find that non-students more often send honest messages. Students send more honest messages when their initial endowment increases, whereas the opposite holds for non-students. If the initial wealth of the opponent increases, students react by sending muss less honest messages. In contrast, the non-student chess players did not change their behavior in this situation. Interestingly, students are more likely to trust the opponent's message. Both, the students and non-students, a much less likely to trust others when their own endowment increases. Thus, we can conclude that there is no clear evidence of whether students or non-students behave more pro-socially.

Replication studies must show whether our findings are artifacts or systematical. For example, it is an open question whether order effects influenced our results. In addition, other levels of the initial endowment (we used €10) and the emotions which are associated with such differences should be analyzed. Moreover, further studies are required if the findings are robust if the games are played for multiple rounds. For example [45], model deception as a multi-period bargaining process in which in an early stage a relationship is established with the victim to exploit it in a later stage.

## Supporting information

**S1 File.**
(PDF)

**S1 Appendix.**
(DOCX)

## Author Contributions

**Conceptualization:** Sven Grüner, Ilia Khassine.

**Data curation:** Sven Grüner.

**Formal analysis:** Sven Grüner, Ilia Khassine.

**Funding acquisition:** Sven Grüner.

**Investigation:** Sven Grüner, Ilia Khassine.

**Methodology:** Sven Grüner, Ilia Khassine.

**Project administration:** Sven Grüner, Ilia Khassine.

**Supervision:** Sven Grüner.

**Validation:** Sven Grüner, Ilia Khassine.

**Visualization:** Sven Grüner, Ilia Khassine.

**Writing – original draft:** Sven Grüner, Ilia Khassine.

**Writing – review & editing:** Sven Grüner, Ilia Khassine.

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
