## [Decision Letter · Decision Letter 0]

1 Oct 2021

PONE-D-21-22531Is there a link between endowment inequality and deception? – An analysis of students and chess playersPLOS ONE

Dear Dr. Grüner,

Thank you for submitting your manuscript to PLOS ONE. After careful consideration, we feel that it has merit but does not fully meet PLOS ONE’s publication criteria as it currently stands. Therefore, we invite you to submit a revised version of the manuscript that addresses the points raised during the review process.

Finally I was able to get the second reviewer. He could not use the PLOS platform but sent the report directly to me. I have attached his file to this message.  Well, both reports are mild, not negative not enthusiastic. As you can see both reviewers share the same impression: they feel that the paper needs to place in the literature, a lot of clarifications and re-writing in order to be a serious contribution and besides that, to be of interest for a wide audience (as PLoS ONE). Personally I feel that most of these amends are doable... so I dont see any reason for not asking for a revision. Please note that I will send the paper back to very same referees. Please submit your revised manuscript by Nov 15 2021 11:59PM. If you will need more time than this to complete your revisions, please reply to this message or contact the journal office at plosone@plos.org. Please include the following items when submitting your revised manuscript:A rebuttal letter that responds to each point raised by the academic editor and reviewer(s). You should upload this letter as a separate file labeled 'Response to Reviewers'.A marked-up copy of your manuscript that highlights changes made to the original version. You should upload this as a separate file labeled 'Revised Manuscript with Track Changes'.An unmarked version of your revised paper without tracked changes. You should upload this as a separate file labeled 'Manuscript'.

We look forward to receiving your revised manuscript.

Kind regards,

Pablo Brañas-Garza, PhD Economics

Academic Editor

PLOS ONE

Journal Requirements:

2. Please update your submission to use the PLOS LaTeX template. The template and more information on our requirements for LaTeX submissions can be found at http://journals.plos.org/plosone/s/latex

3. We note that you have stated that you will provide repository information for your data at acceptance. Should your manuscript be accepted for publication, we will hold it until you provide the relevant accession numbers or DOIs necessary to access your data. If you wish to make changes to your Data Availability statement, please describe these changes in your cover letter and we will update your Data Availability statement to reflect the information you provide

4.  Please include your full ethics statement in the ‘Methods’ section of your manuscript file. In your statement, please include the full name of the IRB or ethics committee who approved or waived your study, as well as whether or not you obtained informed written or verbal consent. If consent was waived for your study, please include this information in your statement as well

Reviewers' comments:

Reviewer's Responses to Questions

**Comments to the Author**

1. Is the manuscript technically sound, and do the data support the conclusions?

Reviewer #1: Yes

2. Has the statistical analysis been performed appropriately and rigorously? 

Reviewer #1: Yes

3. Have the authors made all data underlying the findings in their manuscript fully available?

Reviewer #1: Yes

4. Is the manuscript presented in an intelligible fashion and written in standard English?

Reviewer #1: Yes

5. Review Comments to the Author

Reviewer #1: Summary

The aim of the paper is to better understand the interplay between inequality and deception. To study it they run an experiment using a sender-receiver game where initial endowment of participants vary. Additionally, they compare students and chess players. They find that chess players are more honest, that students trust the message send by senders more often and that initial endowment affects different populations in different ways.

Overall evaluation

In my opinion the paper is interesting and answers an important question in the literature. In spite of this, I have the feeling that it has some flaws. First, the objective of the paper and its importance in the literature should be clarified. Second the design should be clearer for the reader and finally I would clarify the hypothesis and results. You can see all my comments below.

Main comments

- I think that you can further improve the abstract by simplifying the explanation about the basic game and enhancing your differences with it and what you add to the literature, by for example explaining what type of endowment variations you introduce.

- I would mention that your main interest is to study lying to another subject rather than to yourself and that is why you choose the mentioned task instead of the die experiment (Shalvi et al. 2012; Fischbacher and Follmi Heusi, 2013; Charness et al. 2019) or a real effort task (Mazar et al. 2008; Groulleau et al. 2016).

- There are very few experiments in deception with non-students. One example is Utikal and Fischbacher (2013). They use a different task but find differences between the behavior of students and a very specific population, nuns. This comparison is therefore similar to yours.

- Chess players are used to two people game but they are also a population always playing a game where dishonesty if highly punished and socially non-acceptable. This can also affect their behavior. You need to further justify your choice of non-student population. Do you have any information on income? It may be the case that the payoff is less important for adult chess players than for college students.

- I would explain a little bit better the design in the introduction part so that the reader knows what you mean by a variation of the sender-receiver game.

- I have issues following the design. Do they decide the message for both type of payoffs? Please justify why you chose those endowments. Option A (not cheating) is always more efficient but more in situation 1 than 2 for example. When are different endowments announced and how? Because since there is not an effort task it seems that the initial endowment is due to luck which can affect the entitlement over this amount. I would rename the treatments to something more intuitive so that we do not mix them. For example, Poor/rich sender or something similar.

- If they choose their option for both type of situations it may be the case that you have order effects. Did you control for this?

- I would state hypotheses instead of the current way you present your expected results. It is now a little bit confusing to follow this section. This would also make easier to follow the rest of the sections.

- You have regressions controlling for senders who believe receiver follow their message. Do you obtain the same if you leave out strategic senders?

Other comments

- I think that mentioning the main results in the introduction could add to the paper.

- You probably mean payed instead of played in the sentence below. Otherwise I do not understand the meaning of the sentence.

“Subjects in the role of player 1 had to decide in two situations. We flipped a coin (i.e., p = 0.5) to determine which of the two decision situations were to be played (i.e., random lottery payment technique)”

- I would not introduce the formula of Fehr and Schmidt (1999) if I am not using it again in the rest of the paper. It can be misleading.

- I do not see the point of having part 4.2.

- You need to further explain why you ask senders about their expectation regarding what receivers will do (Sutter, 2009). As it is now I am not aware that you controlled for this until the results section.

- It may be the case that when receivers have a low endowment they do not trust senders because they may believe that they will try to even the payoffs.

- You find some gender differences in behavior. There are multiple papers in gender differences in dishonesty finding different results (Dreber and Johannesson, 2008; Gylfason et al., 2013; Ezquerra et al. 2018). It could add to the paper to mention and discuss what you find related to gender.

References

Charness, G., Blanco-Jimenez, C., Ezquerra, L., & Rodriguez-Lara, I. (2019). Cheating, incentives, and money manipulation. Experimental Economics, 22(1), 155-177.

Gylfason, H. F., Arnardottir, A. A., & Kristinsson, K. (2013). More on gender differences in lying. Economics Letters, 119(1), 94-96.

Dreber, A., & Johannesson, M. (2008). Gender differences in deception. Economics Letters, 99(1), 197-199.

Ezquerra, L., Kolev, G. I., & Rodriguez-Lara, I. (2018). Gender differences in cheating: Loss vs. gain framing. Economics Letters, 163, 46-49.

Fischbacher, U., & Föllmi-Heusi, F. (2013). Lies in disguise—an experimental study on cheating. Journal of the European Economic Association, 11(3), 525-547.

Grolleau, G., Kocher, M. G., & Sutan, A. (2016). Cheating and loss aversion: Do people cheat more to avoid a loss?. Management Science, 62(12), 3428-3438.

Mazar, N., Amir, O., & Ariely, D. (2008). The dishonesty of honest people: A theory of self-concept maintenance. Journal of marketing research, 45(6), 633-644.

Shalvi, S., Eldar, O., & Bereby-Meyer, Y. (2012). Honesty requires time (and lack of justifications). Psychological science, 23(10), 1264-1270.

Utikal, V., & Fischbacher, U. (2013). Disadvantageous lies in individual decisions. Journal of Economic Behavior & Organization, 85, 108-111.

6. PLOS authors have the option to publish the peer review history of their article (what does this mean?). If published, this will include your full peer review and any attached files.

Reviewer #1: No

---

## [Author Response · Author response to Decision Letter 0]

8 Nov 2021

PONE-D-21-22531

Is there a link between endowment inequality and deception? – An analysis of students and chess players

Reply to the reviewers

We very much appreciate the reviewers’ thorough reading of our manuscript as well as their very thoughtful comments. In the following, we have listed the reviewers’ comments along with our replies and the cross reference to the changes implemented in the manuscript. 

1 Reviewer 1

- I think that you can further improve the abstract by simplifying the explanation about the basic game and enhancing your differences with it and what you add to the literature, by for example explaining what type of endowment variations you introduce.

The basic design is described as follows:

“This paper investigates experimentally the relationship between inequality in endowment and deception. Our basic design is adopted from Gneezy (2005): two players interact in a deception game. It is common knowledge that player 1 has private information about the payoffs for both players of two alternative actions. Player 1 sends a message to player 2, indicating which alternative putatively will end up in a higher payoff for player 2. The message, which can either be true or false, does not affect the payoffs of the players. Player 2 has no information about the payoffs. However, player 2 selects one of the two alternatives A or B, which is payoff-relevant for both players.” 

The added value to the literature is explained:

“Our paper adds value to the literature by extending Gneezy (2005) in two elements. First, we systematically vary the initial endowment of the players 1 and 2 (common knowledge to both of them). Second, we do not limit ourselves to the standard population of university students but also recruit chess players that are not enrolled in any degree program. Doing so, we want to find out if our results remain robust over a non-standard subject population which is known to be experienced to some extent in strategic interactions.” 

We would like to refrain from providing details about the level of the initial endowment because this is not that important to understand the paper itself.

The findings are described as follows:

“Our main findings are: (i) non-students behave more honestly than students, (ii) students are more likely to trust the opponent’s message, and (iii) students and non-students behave differently to variation in initial endowment.” 

- I would mention that your main interest is to study lying to another subject rather than to yourself and that is why you choose the mentioned task instead of the die experiment (Shalvi et al. 2012; Fischbacher and Follmi Heusi, 2013; Charness et al. 2019) or a real effort task (Mazar et al. 2008; Groulleau et al. 2016).

We added your point to the paper and now say:

“To study lying to other people (instead of lying to yourself in which the die experiment or a real effort task are quite common; e.g. Grolleau et al. 2016; Charness et al. 2019), our paper adopts the basic design of Gneezy’s (2005) two-player cheap talk sender-receiver game.”

Please notice that we would like to pick the two most recent references. More importantly, we refrain from the Mazar reference because Ariely was involved. We have some serious doubts about this reference. Recently, there was some public attention to a paper of Ariely and doubts about data generation and analysis. For example, a brief discussion about this can be found here: https://datacolada.org/98. It is too risky for us to include Ariely here because we cannot fully replicate the study to find out if it is based on valid research.

- There are very few experiments in deception with non-students. One example is Utikal and Fischbacher (2013). They use a different task but find differences between the behavior of students and a very specific population, nuns. This comparison is therefore similar to yours.

We added the Utikal and Fischbacher (2013) as one of the few experiments with systematic subject pool comparisons to our paper. We now say: 

“Subject pool differences have been rarely analyzed in the realm of deception. One important exception is Utikal and Fischbacher (2013). They compare students and nuns in a lying experiment and find differ-ences between the subject pools (most notably, nuns are lying to their disadvantage in individual decision problems).”

- Chess players are used to two people game but they are also a population always playing a game where dishonesty if highly punished and socially non-acceptable. This can also affect their behavior. You need to further justify your choice of non-student population. Do you have any information on income? It may be the case that the payoff is less important for adult chess players than for college students.

We are not sure if chess players are playing a game where dishonesty is highly punished. In chess, people do not want to show their real emotions. For example, if they have a worse position, some players pretend to be completely disinterested in the game. So that the opponent thinks that it must be an easy task to win the game.

Our analysis comprises comparing mean values between the different subject pools and, in a further step, regression analysis. In the regression analysis, we use net income as a control variable. So, yes, you are right that the stakes might be a point for differences but we control for that econometrically.

We agree with you that we should stronger motivate the use of chess players. In the revised version we now consider identity economics as theoretical background in association with the “competition spirit” of chess. We added the following sentences: 

“Moreover, by reminding them that we are looking for chess players, there might be some weak form of priming (Cohn and Maréchal 2016). As Akerlof and Kranton (2000) have pointed out people have different identities. The subjects may be in a competitive mood when they are reminded of their identity as a chess player (in chess there are only 3 outcomes: draw, win or lose).” 

- I would explain a little bit better the design in the introduction part so that the reader knows what you mean by a variation of the sender-receiver game.

We have added a brief note about the variation of the initial endowment. We now say:

“In our extended Gneezy (2005) design, we provide either player 1 or 2 with an initial endowment of €10 in the treatment conditions.”

We would like to keep it in a brief way because the design is explained in detail in the section after the Introduction.

- I have issues following the design. Do they decide the message for both type of payoffs? Please justify why you chose those endowments. Option A (not cheating) is always more efficient but more in situation 1 than 2 for example. When are different endowments announced and how? Because since there is not an effort task it seems that the initial endowment is due to luck which can affect the entitlement over this amount. I would rename the treatments to something more intuitive so that we do not mix them. For example, Poor/rich sender or something similar.

Yes, the subjects decide for both situations. After allocating the subjects to one treatment, they are informed about the initial endowment (and the initial endowment of the opponent). We have attached the experimental instructions to the manuscript for transparency. 

“We extend Gneezy (2005) by considering systematic variations of the initial endowment. Altogether, we examine 3 scenarios (1 reference scenario and 2 treatment scenarios; Table 2). The experimental subjects were randomly assigned to one scenario only. To increase the statistical power of our analysis, subjects had to respond to both situations within one endowment scenario (between-subjects study design). We refrain from using a within-subjects study design (each experimental subject makes decisions in all endowment scenarios), as the subjects may activate different emotions in the treatments. Player 1 and player 2 know that the initial endowment in the respective treatments and are aware that it is common knowledge to both players.”

We are not sure about using label such as “rich” or “poor” because it is too extreme (it was only about €10). However, we tried to find some places in the paper to add reminders about the meaning of the treatments. For example, in Table 3 we added the description of the treatments. Another example is the paragraph above of Table 6 where the treatment conditions were also defined again.

The level of the initial endowment: the short answer is that this is an empirical / experimental question if our level was adequate or not. We thought that even small differences in the initial endowment might infer emotions / inequity aversion. This should be addressed in further research. Therefore, we write in the Conclusion section:

“For example, other levels of the initial endowment (we used €10) and the emotions which are associated with such differences should be analyzed.”

- If they choose their option for both type of situations it may be the case that you have order effects. Did you control for this?

In Table 1, we added the following note:

“We presented situation 1 to the subjects first. We cannot exclude the possibility of order effects, which has to be analyzed in follow-up studies.”

- I would state hypotheses instead of the current way you present your expected results. It is now a little bit confusing to follow this section. This would also make easier to follow the rest of the sections.

We can understand your point by we pre-registered research questions and would like to follow the pre-registration as close as possible to avoid confusion of why we deviate from it. The basic idea was that often null hypotheses are formulated (tests against a non-realistic null hypothesis; cf. John List in a paper on the market of publications) – we did not want to follow this practice.

- You have regressions controlling for senders who believe receiver follow their message. Do you obtain the same if you leave out strategic senders?

I think this is an important topic. In Table 7a and 7b, we provide regressions on both versions (with and without expectations of whether the other person follows/beliefs or not). Direct comparisons are possible and briefly explained in the manuscript.

Other comments

- I think that mentioning the main results in the introduction could add to the paper.

We spent a plenty of time thinking of the pros and cons of adding the results to the Introduction. Yes, there are some journals that follow this practice. We would like to stick with the old version because we think that the reader should first better understand the experimental design. The results depend so much on details in experimental economics. Therefore, we think that talking about the findings could be one version but does not add that much value to the paper.

- You probably mean payed instead of played in the sentence below. Otherwise I do not understand the meaning of the sentence.

“Subjects in the role of player 1 had to decide in two situations. We flipped a coin (i.e., p = 0.5) to determine which of the two decision situations were to be played (i.e., random lottery payment technique)”

Yes, this typo is rather is a rather serious one. Thank you for pointing it out and reading it that carefully. The typo has been corrected (we now write “paid”).

- I would not introduce the formula of Fehr and Schmidt (1999) if I am not using it again in the rest of the paper. It can be misleading.

By providing a formula, we intended to illustrate the components that we are interested in. Some people prefer thinking in form of text, others are better in understanding formulas. So both versions might provide something for both.

- I do not see the point of having part 4.2.

Recently, we experienced that many reviewers argued that nothing can be learned from findings p > 0.05. That’s why we say: “Don’t believe that an association or effect is absent just because it was not statistically significant.” We say this because we have many results p>0.05.

- You need to further explain why you ask senders about their expectation regarding what receivers will do (Sutter, 2009). As it is now I am not aware that you controlled for this until the results section.

In Section 4.1, we now say:

“It is important to consider subjects’ expectations about the likely behavior of others because both preferences and beliefs matter (which is similar to public goods experiments, for example).” 

Thank you for this point. We think that making this statement definitely improves clarity of the paper.

- It may be the case that when receivers have a low endowment they do not trust senders because they may believe that they will try to even the payoffs.

To be honest, we were really surprised how complicated such a seemingly simple experiment can be. Our guess was that player 2, who does not have information on the payoff, is more likely to follow player 1 if player 2 thinks that the other player wants to level the payoffs (e.g. due to declining inequality). Here, the formula helps to play a little bit with the determinants.

- You find some gender differences in behavior. There are multiple papers in gender differences in dishonesty finding different results (Dreber and Johannesson, 2008; Gylfason et al., 2013; Ezquerra et al. 2018). It could add to the paper to mention and discuss what you find related to gender.

Regarding gender (which was not pre-registered) we have mixed effects. In the first situation, it seems that women are more likely to send honest messages (which is in line with Dreber and Johannesson 2008). However, in the other situation, the sign of women is negative, but the p-values are very high, so we should be cautious about preliminary findings. Thinking in terms of significant/not-significant, as the other paper do, we would say that we “do not find differences” in situation 2, which is in line with Gylfasion et al. 2013 and Ezquerra et al. 2018. But further research is necessary. However, gender was merely a control variable, not our focus of interest.

 

2 Reviewer 2

From the abstract and the introduction of your article, the reader could easily identify two goals of the article: contribute to the understanding of the relationship between inequalities and deception and improve the external validity of the results in experimental economics. While I think that the experimental design provides us with one step forward to reaching our first goal, I cannot stop asking, "why should we particularly care about the deceptive behavior of chess players? This question was raised by Frechette (2011) in the discussion when he mentions: 

"It also raises the question of what is the group of interest? Who are the agents that are supposed to be represented in those models being tested?"

Thus, how does the selection and understanding of the differences in chess players' behavior, in this case, improve the external validity of your results? I believe that you should develop on answering these questions in your introduction and extending your literature review on different samples of experimental subjects and the implications of observing differences in the behavior of these types in experimental games. Good examples to follow are one given by Abbink and Rochenbak (2006), Alevy et al. (2009), where they explain why understanding the behavior of financial advisors in a cascade game is crucial to improve our understanding of cascades. 

We agree with you that we should stronger motivate the use of chess players. In the revised version we now consider identity economics (Akerlof / Kranton) as theoretical background in association with the “competition spirit” of chess. We motivate the recruitment of chess players as follows:

“Chess players seem to be interesting for our experiment because they have training in strategic interactions. While playing chess, individuals not only have to think about objectively good moves but also form expectations about how the opponent might react to them. For example, an objectively perfect move could work out poorly in practice if it leads to variations where the opponent is an expert in. Similar to our experiment, in chess usually two people play against each other. Moreover, by reminding them that we are looking for chess players, there might be some weak form of priming (Cohn and Maréchal 2016). As Akerlof and Kranton (2000) have pointed out people have different identities. The subjects may be in a competitive mood when they are reminded of their identity as a chess player (in chess there are only 3 outcomes: draw, win or lose).”

Furthermore, the introduction of two types of subjects increases your hypothesis. Thus, you should provide power estimations that drive your experimental design and the choice of your number of observations per treatment and correct when needed for multiple hypothesis testing. I don't know if I failed to discover this analysis and values, but it is very important to perform these tasks even more when we have multiple populations.

Thanks for this comment. We now provide reasons for our sample size. We say:

“The size of the sample was primarily driven by budget constraints.”

However, ex post sample size calculations are seen very critically by many scientists. One of the authors has written a paper on power calculations. Just to provide one example: 

https://blogs.worldbank.org/impactevaluations/why-ex-post-power-using-estimated-effect-sizes-bad-ex-post-mde-not

As a consequence we would like stick with the budget constraint argument (which was the true reason for our sample size).

When there are multiple subject pools and, in turn, multiple interaction terms, researchers often face multicollinearity. Since this does not influence marginal effects and signs (only makes p-values high), it is not that much a problem if our focus is on signs and coefficients. Multiple hypothesis testing is always a topic. We add this point to the comment on p-values (Section 4.2):

“Notice, in our sample, there are many interaction terms (which deflates p-values) and several variables we looked at (“multiple testing”, which inflates p-values). Therefore, p-values should be cautiously interpreted. The signs and strength of evidence in terms of marginal effects or differences in mean are more meaningful than just looking at p-values.”

While reading the article, I found some style and expression habits that I believe a difficult reader understands your work. For example, the article has several instances where the terms such as Player 1 and Player 2 are used to label tables. Then the reader finds that some other sentences are labeled as sender and receiver, which are used, for example, to name sub-sections of the paper leads the reader in a constant loop of mapping between several different terms to understand the text of your article. It would be an improvement in terms of the article's readability if you could reduce the diversity of terms or use them more systematically. N additional concept that appears to be misleading to any readers is your definition of "Market implications," this Is a two-person game with a context that Vernon Smith classifying inside the "personal exchange" (Smith 2009). There is no market because subjects are not exchanging a good or service, and there is no price or fee. I think that to be more traditional with the economist interpretation of the market and 2 by 2 experimental games; your article will benefit

Thank you. We improved the readability of the paper by not jumping between the terms. Moreover, if the terms player 1 or player 2 were used, we attached the label “sender” and “receiver” if there might be some confusion. Most importantly, in the description of the variables of the paper (Table 4), we now say “player 1 (sender”) instead of only “player 1”, for example. The same has been done for player 2. Similar changes have been made in Table 6 and 8, for example, too.

Following your advice, we now refrain from using the term “market”. Thus, there are some relabeling in the sections 3.3 (research questions), 4.1 (approach to data analysis), and 6.3. (data analysis). We now speak of bargaining outcome.

I believe that the revision of related literature should be expanded, and the text should highlight how alternative theoretical frameworks and previous experiments related to your design and results. For example, little is said about how endowment inequalities affect behavior in other experimental games while there is ample evidence of endowment inequalities among others affecting behavior in trust games such as the one given by: Anderson et al. 2006, Ciriolo 2007, Lei & Vesely 2010, Xiao & Bicchieri 2010, Smith 2011, Brülhart & Usunier 2012, Hargreaves-Heap et al. 2013, Calabuig et al. 2016, Rodriguez-Lara 2018, Bejarano et al. 2018, 2021. Similarly, due to the lack of revision of different theoretical foundations of the relationship between inequality and deception, it appears that there is only one theoretical framework to base your hypothesis is only based on Fehr and Schimdt (1999). A set of new theories of deception Ettinger and Jehiel (2010) or even some interpretations of Mazar et al., 2008 could also inform you, the reader, and other researchers of how your hypothesis and results are in place within the larger literature on deception.

In our revision, we consider some studies which address inequality. This helps to better motivate the research topic of our paper. We say: 

“The topic of inequality has been tackled in the experimental literature with mixed findings. For example, in the realm of trust games, Bejarano et al. (2021) found evidence for inequality to matter, whereas Rodriguez-Lara (2018) does not find evidence for inequality aversion.”

Thank you for pointing to the Ettinger and Jehiel (2010) reference. We didn’t know this paper. But it is very interesting. Especially, we like the bargaining idea. Therefore, we added the following sentences in the last section of the paper: 

“Moreover, further studies are required if the findings are robust if the games are played for multiple rounds. For example, Ettinger and Jehiel (2010) model deception as a multi-period bargaining process in which in an early stage a relationship is established with the victim to exploit it in a later stage.” 

However, we would refrain from the Mazar reference because Ariely was involved. We have some serious doubts about this reference. Recently, there was some public attention to a paper of Ariely and doubts about data generation and analysis. For example, a brief discussion about this can be found here: https://datacolada.org/98. It is too risky for us to include Ariely here because we cannot fully replicate the study to find out if it is based on valid research.

Finally, I consider that your statement regarding p-values and econometric analysis is understandable regarding the analysis. And even after carefully reading your footnote 4" where you stated: "we refrain from using asterisks and provide complete p-values instead, we reduce the number of econometric specifications than originally intended and pre-registered due to space restrictions. For a comprehensive overview of the specification, see the Appendix." 

I cannot stop thinking that there is some inconsistency in your analytical approach. First, you stated reduced form regression expressions and present proportions and the outcomes of the statistical tests between proportions. Then you present tables with marginal effects of "only" one reduce form regression, the logit. 

But you do not talk about the robustness of the effects to different specifications or the need of clustering the errors, and you present the p-values in these tables. Thus this inconsistency claiming that p-values are not the only important factor and then presenting tables with p-values, but abstaining from defining a clear analytical criterion of analysis difficult readers comprehension of your results. Finally, I can not stop thinking that you do not have many significant results in statistical and non-statistical meaning. Worse than not having significant results is not clearly stating the results or lack of them. I think that the readers would appreciate it if you could clarify the criteria for your analysis. Even if you produce the traditional analysis, show the significance of these results and state your position within the body of the text in the results and discussion section in a more ordinated fashion.

We provide the results for all pre-registered specifications in the paper and some supporting material (robustness checks in the form of correlations, in the Appendix). For example, the zero order correlations are important as a robustness check. There might be some mediator effects in the course of the regression analysis. Therefore, it is important to provide both the correlation analysis (in the Appendix) and the regressions in the manuscript. We only speak of the correlation analysis if there are deviations from the regression analysis. Otherwise it is the same content and only serves as a robustness check.

There are also robustness checks in the regression analysis: for example, there are specifications where the variable expectations has been included and, in another regression, was included. This helps to find out whether the results are “stable.” 

We have rewritten the statement on p-values in Section 4.2 and now stronger say, where the focus of the analysis is:

“There is an intensive debate and discussion on how to use and interpret p-values (Wasserstein et al. 2019). Since this article is not the appropriate place to pursue the discussion in detail, we want to communicate only a few thoughts. While in the past it was quite common to focus on “statistically significant” results, the dichotomy of significant/non-significant is increasingly viewed critically. For example, Wasserstein et al. (2019: 1, 2): argue “Don’t believe that an association or effect is absent just because it was not statistically significant.” or “In sum, “statistically significant”—don’t say it and don’t use it.

Notice, in our sample, there are many interaction terms (which deflates p-values) and several variables we looked at (“multiple testing”, which inflates p-values). Therefore, p-values should be cautiously interpreted. The signs and strength of evidence in terms of marginal effects or differences in mean are more meaningful than just looking at p-values.”

We do not only provide logit-regressions: there are mean-values, correlations of the regression analysis as well as robustness checks (zero-order correlations, in the Appendix)

---

## [Decision Letter · Decision Letter 1]

26 Nov 2021

PONE-D-21-22531R1Is there a link between endowment inequality and deception? – An analysis of students and chess playersPLOS ONE

Dear Dr. Grüner,

Thank you for submitting your manuscript to PLOS ONE. After careful consideration, we feel that it has merit but does not fully meet PLOS ONE’s publication criteria as it currently stands. Therefore, we invite you to submit a revised version of the manuscript that addresses the points raised during the review process.

I have selected "minor revision" because there is no much work to do but you must carefully address  all the minor points raised by the reviewer. ==============================

We look forward to receiving your revised manuscript.

Kind regards,

Natalia Jiménez, Ph.D. Economics

Academic Editor

PLOS ONE

Journal Requirements:

Additional Editor Comments:

Dear authors,

It seems that now the reviewer is quite satisfied with how you have addressed his/her concerns with a few exceptions. You should carefully address the few points raised by the referee, especially the one that you didn´t answer in the previous revision (the one in quotation marks).

Reviewers' comments:

Reviewer's Responses to Questions

**Comments to the Author**

1. If the authors have adequately addressed your comments raised in a previous round of review and you feel that this manuscript is now acceptable for publication, you may indicate that here to bypass the “Comments to the Author” section, enter your conflict of interest statement in the “Confidential to Editor” section, and submit your "Accept" recommendation.

Reviewer #1: (No Response)

2. Is the manuscript technically sound, and do the data support the conclusions?

Reviewer #1: Yes

3. Has the statistical analysis been performed appropriately and rigorously? 

Reviewer #1: Yes

4. Have the authors made all data underlying the findings in their manuscript fully available?

Reviewer #1: (No Response)

5. Is the manuscript presented in an intelligible fashion and written in standard English?

Reviewer #1: (No Response)

6. Review Comments to the Author

Reviewer #1: There is a big change from the previous version to this one and most of my comments have been addressed. I have some small comments for the authors though.

Main comments

- You need to further develop the possibility of having different identities. It should be added to the conclusions as a potential explanation for different results. Regarding income how it was calculated should be explained.

- Regarding order effects I think that this is a major drawback and as such you should mention it in the conclusions rather than in a footnote.

- Even though you do not directly answer the hypothesis I think you should clarify your result section so that is easier for the reader to understand the main results obtained from your study.

Other comments

- I think that mentioning the main results in the introduction would add to the paper but I respect that you think otherwise.

- Regarding the formula Fehr and Schmidt (1999) I would comment the purpose of having it there and that you are not using it further so that it is not misleading.

- I feel that you did not answer to my comment: “It may be the case that when receivers have a low endowment they do not trust senders because they may believe that they will try to even the payoffs”. I would add this as a possible explanation.

- It was difficult to understand this sentence of the introduction “Player 1 sends a message to player 2, which of the two options is supposedly financially advantageous for player 2. “ I think you cound add something like stating, explaining…after the comma.

- Clarify in the introduction if the initial endowment is salient for all players.

7. PLOS authors have the option to publish the peer review history of their article (what does this mean?). If published, this will include your full peer review and any attached files.

Reviewer #1: No

---

## [Author Response · Author response to Decision Letter 1]

26 Nov 2021

PONE-D-21-22531R1

Is there a link between endowment inequality and deception? – An analysis of students and chess players

PLOS ONE

Many thanks for your comments. Please find attached our response along with the changes we have made in the manuscript.

- You need to further develop the possibility of having different identities. It should be added to the conclusions as a potential explanation for different results. Regarding income how it was calculated should be explained.

Thanks for this point. We added the possible reason (different identities) to the Conclusion Section: “Reminding the subjects of their identity might be a driver for differences.”

The variables are explained in Table 4. For example, we used several categories to collect data on net income: “less than €750 (=1), €750 up to less than €1,500 (=2), €1,500 up to less than €2,000 (=3), €2,000 up to less than €2,500 (=4), €2,500 up to less than €3,000 (=5), more than €3,000 (=6)”

- Regarding order effects I think that this is a major drawback and as such you should mention it in the conclusions rather than in a footnote.

We now say it not only in the experimental design (we think this should also be said there) but also in the Conclusion Section talking about limitations. 

“Replication studies must show whether our findings are artifacts or systematical. For example, it is an open question whether order effects influenced our results.”

- Even though you do not directly answer the hypothesis I think you should clarify your result section so that is easier for the reader to understand the main results obtained from your study.

Yes, you are right. Results is a better description than “Data analysis”. Section 6 is not labeled as Results.

Other comments

- I think that mentioning the main results in the introduction would add to the paper but I respect that you think otherwise.

Thank you. Yes, there are different ways in the literature. �

- Regarding the formula Fehr and Schmidt (1999) I would comment the purpose of having it there and that you are not using it further so that it is not misleading.

We now introduce the formula as follows:

“For illustration purposes, this will be presented formally (although the formula will not be used later in the paper):”

- I feel that you did not answer to my comment: “It may be the case that when receivers have a low endowment they do not trust senders because they may believe that they will try to even the payoffs”. I would add this as a possible explanation.

We think that this point is somewhat related to the definition issue we have written in the paper. This might be the case that we did not consider it in the revision (lost in thoughts). This was also in the old version of the paper: “To keep it simple, we stick to the terminology of Gneezy (2005) in this paper. However, we would like to point out that this dichotomy (honest vs. deception) could also be seen critically. For example, as Sutter (2009) correctly points out honest messages could also be classified as deception if one has the expectation that the receiver does not follow the message.”

And now, we added the following sentence:

“It should also be mentioned that receivers, who have less initial endowment than their opponents, might not trust the senders because the former could believe that the latter may try to even the payoffs as a fairness criterion.”

- It was difficult to understand this sentence of the introduction “Player 1 sends a message to player 2, which of the two options is supposedly financially advantageous for player 2. “ I think you cound add something like stating, explaining…after the comma.

We think that we missed a word in our sentence and made it hard to read. We added “indicating” after the comma (but stating was also fine to us). Thank you.

- Clarify in the introduction if the initial endowment is salient for all players.

In the Introduction, we describe the basic design along with the treatment conditions where we systematically vary in the endowment:

“In our extended Gneezy (2005) design, we provide either player 1 or 2 with an initial endowment of €10 in the treatment conditions. In accordance with Fehr and Schmidt (1999), we distinguish between monetary advantageous inequality and monetary disadvantageous inequality.”

---

## [Decision Letter · Decision Letter 2]

17 Dec 2021

Is there a link between endowment inequality and deception? – An analysis of students and chess players

PONE-D-21-22531R2

Dear Dr. Grüner,

We’re pleased to inform you that your manuscript has been judged scientifically suitable for publication and will be formally accepted for publication once it meets all outstanding technical requirements.

Kind regards,

Natalia Jiménez, Ph.D. Economics

Academic Editor

PLOS ONE

Additional Editor Comments (optional):

Reviewers' comments:

Reviewer's Responses to Questions

**Comments to the Author**

1. If the authors have adequately addressed your comments raised in a previous round of review and you feel that this manuscript is now acceptable for publication, you may indicate that here to bypass the “Comments to the Author” section, enter your conflict of interest statement in the “Confidential to Editor” section, and submit your "Accept" recommendation.

Reviewer #1: All comments have been addressed

2. Is the manuscript technically sound, and do the data support the conclusions?

Reviewer #1: Yes

3. Has the statistical analysis been performed appropriately and rigorously? 

Reviewer #1: Yes

4. Have the authors made all data underlying the findings in their manuscript fully available?

Reviewer #1: (No Response)

5. Is the manuscript presented in an intelligible fashion and written in standard English?

Reviewer #1: Yes

6. Review Comments to the Author

Reviewer #1: (No Response)

7. PLOS authors have the option to publish the peer review history of their article (what does this mean?). If published, this will include your full peer review and any attached files.

Reviewer #1: No

---

## [Editor Report · Acceptance letter]

12 Jan 2022

PONE-D-21-22531R2 

Is there a link between endowment inequality and deception? – An analysis of students and chess players 

Dear Dr. Grüner:

I'm pleased to inform you that your manuscript has been deemed suitable for publication in PLOS ONE. Congratulations! Your manuscript is now with our production department. 

Kind regards, 

on behalf of

Dr. Natalia Jiménez 

Academic Editor

PLOS ONE